# The Combination of CD300c Antibody with PD-1 Blockade Suppresses Tumor Growth and Metastasis by Remodeling the Tumor Microenvironment in Triple-Negative Breast Cancer

**DOI:** 10.3390/ijms26115045

**Published:** 2025-05-23

**Authors:** Soyoung Kim, Ik-Hwan Han, Suin Lee, DaeHwan Park, Hyunju Lee, Jongyeob Kim, Joon Kim, Jae-Won Jeon, Hyunsu Bae

**Affiliations:** 1Department of Science in Korean Medicine, College of Korean Medicine, Kyung Hee University, 26 Kyungheedae-ro, Dongdaemun-gu, Seoul 02447, Republic of Korea; samanda0@nate.com (S.K.); hihan@khu.ac.kr (I.-H.H.); xkdla992001@naver.com (D.P.); sallyhyunju@naver.com (H.L.); 2CentricsBio Inc., 28 Beobwon-ro 11-gil, Songpa-gu, Seoul 05836, Republic of Korea; silee@centricsbio.com (S.L.); jongyeob.kim@centricsbio.com (J.K.); joon.kim@centricsbio.com (J.K.)

**Keywords:** breast cancer, anti-PD-1, CD300c, CL7, tumor associated macrophage, combination therapy

## Abstract

Triple-negative breast cancer (TNBC) is an aggressive cancer characterized by a high risk of recurrence, invasiveness, metastatic potential, and poor prognosis. Tumor-associated macrophages (TAMs), particularly M2-like TAMs, contribute to TNBC progression by promoting an immunosuppressive tumor microenvironment (TME), highlighting the need for TME remodeling. This study aimed to evaluate the therapeutic efficacy of co-administering CL7, a CD300c monoclonal antibody that induces M1 macrophage polarization, and anti-PD-1, an immune checkpoint inhibitor, in TNBC. To establish a TNBC model, 4T1 cells were inoculated into the fourth left mammary gland of mice. CL7 and anti-PD-1 were intravenously administered twice a week. Flow cytometry and RT-PCR were performed to assess the immunotherapeutic effects, and lung metastases were evaluated by the Hematoxylin and Eosin staining of lung tissues. Tumor growth was significantly reduced in the combination treatment group (CL7 and anti-PD-1) compared to both the PBS and monotherapy groups. Additionally, the combination treatment increased M1 macrophages and activated CD8+ T and NK cells in the tumor, while significantly suppressing lung metastases. These findings suggest that the combination of CL7 and anti-PD- therapy has the potential to treat TNBC by remodeling the TME.

## 1. Introduction

Triple-negative breast cancer (TNBC) is an aggressive malignancy characterized by the absence of an estrogen receptor (ER) and progesterone receptor (PR), as well as the lack of an overexpression of human epithelial cell growth factor receptor 2 (HER2) [1]. TNBC is known for its rapid tumor growth, which triggers the host immune response and leads to significant lymphocyte-driven inflammation. As the tumor expands, cell adhesion is disrupted, leading to tumor cell migration and invasion into blood and lymphatic vessels, ultimately resulting in metastasis to the lungs, liver, and brain [2].

The tumor microenvironment (TME) is composed of tumor cells, stroma, immune cells, blood vessels, and extracellular matrices [3,4]. TME plays a pivotal role in tumor initiation and progression by regulating key processes such as tumor cell growth, immune evasion, and angiogenesis [5,6]. In breast cancer, TME not only facilitates tumor invasion and metastasis but also significantly influences patient responses to treatment and prognosis.

Cancer immunotherapy aims to enhance the immune system to target and eliminate tumor cells, while also modifying the TME to improve treatment outcomes [7]. Although immunotherapy is considered a promising therapeutic approach in several cancers, monotherapy using immunotherapy is often limited in efficacy, with immune escape and drug resistance emerging as challenges, particularly due to factors inherent within the TME [8]. Immunotherapy can alter the interaction between breast cancer and its microenvironment, thereby enhancing the anti-tumor immune response [9]. Regulating the TME is crucial for improving immunotherapy outcomes in TNBC [8,10,11,12].

In the TME, tumor-associated macrophages (TAMs) exhibit high plasticity, enabling them to differentiate into various phenotypes in response to different TME factors and cytokine stimuli. Specifically, TAMs are classified into anti-tumoral M1 phenotypes and pro-tumoral M2 phenotypes that promote tumor growth and metastasis [13,14]. Understanding the dynamic properties and flexibility of TAMs provides valuable insights for designing treatment strategies by targeting or reprogramming them.

CD300 proteins are expressed on a variety of immune cells, where they are involved in both stimulatory and inhibitory receptor activities [15]. CD300c is expressed on the surface of human monocytes and monocyte-derived cells, including macrophages and dendritic cells. In the previous study, it has been demonstrated that targeting CD300c with the monoclonal antibody CL7 induced M1 macrophage polarization, activated the MAPK and NF-κB signaling pathways, and promoted the secretion of pro-inflammatory cytokines such as TNF-α and IL-1β [16].

Here, we aimed to investigate whether CL7 (CB301) and anti-PD-1 combination treatment affects the reduction of tumor growth and the suppression of lung metastasis in TNBC, by remodeling immune cells in TNBC TME.

## 2. Results

### 2.1. Combination Treatment of CL7 and Anti-PD-1 Synergistically Reduced the Tumor Growth in 4T1 TNBC Model

To evaluate the anticancer effect of CL7 in the 4T1 model, CL7 was administered intraperitoneally (Figure 1A), resulting in a dose-dependent decrease in tumor growth (Figure 1B). To optimize drug efficacy, the administration route and dosage were determined prior to treatment (Figure 1A). In the 4T1 model, intravenous injection induced a more pronounced tumor growth reduction compared to intraperitoneal injection. Furthermore, tumor growth decreased in a dose-dependent manner (Figure 1C). To further investigate the effect of CL7 and anti-PD-1 combination therapy on tumor growth in a TNBC model, we established a TNBC mouse model and administered CL7 and anti-PD-1 (Figure 1D). Compared with the PBS group, the combination treatment significantly reduced both tumor size and weight. Moreover, tumor growth was significantly lower in the combination group than in the monotherapy groups (Figure 1E,F). These findings suggest that combination therapy with CL7 and anti-PD-1 synergistically reduces tumor growth in TNBC.

### 2.2. Combination Treatment of CL7 and Anti-PD-1 Has Effects on the Macrophage Population in the Tumor Microenvironment of TNBC

To determine whether CL7 influences changes in macrophage populations within the TNBC TME, we analyzed the macrophage population using flow cytometry in TNBC tumor tissues (Figure 2A). The combination group showed a significant increase in the M1 macrophage population compared to the PBS and monotherapy groups (Figure 2B,C). No significant difference in the M2 macrophage population was observed in the combination group compared to the PBS and CL7 groups (Figure 2B,D). Additionally, the M1/M2 ratio was significantly higher in the combination group, compared to the other groups (Figure 2E). To assess changes in M1 and M2 macrophages at the mRNA level, RT-PCR was performed. The expression of M1 macrophage-related markers was significantly upregulated in the combination group compared to the other groups (Figure 2F). In contrast, the expression of M2 macrophage-related markers was significantly downregulated in the combination group compared to the PBS group (Figure 2G). Considering these results along with the flow cytometry data, we suggest that the combination treatment of CL7 and anti-PD-1 promotes an increase in M1 macrophages.

### 2.3. Combination Treatment of CL7 and Anti-PD-1 Significantly Enhanced the Activating CD8+ T Cells in the Breast Cancer Tissue

CD8+ T cells mediate antitumor immunity by recognizing and killing tumor cells through MHC class I-restricted antigen presentation, as well as releasing cytotoxic molecules such as Perforin and Granzyme B [17,18]. Furthermore, it has been reported that M2-like macrophages cause the dysregulation of T cell receptor signaling, leading to the inactivation of CD8+ T cells [19]. To investigate the potential of CL7 and anti-PD-1 combination therapy in activating CD8+ T cells in TNBC models, flow cytometry was performed using TNBC tissues (Figure 3A). Although the population of CD8+ T cells did not show a significant difference, it can be seen that the number and percentage of activating CD8+ T cells were significantly increased in the combination group compared to the PBS and monotherapy groups (Figure 3B). According to this result, it is suggested that CL7 and anti-PD-1 synthetically increase activating CD8+ T cells within the TNBC TME.

### 2.4. CL7 and Anti-PD-1 Combination Treatment Promotes NK Cell Activation in the Breast Cancer

To determine the effect of NK cells in TNBC following combination therapy with CL7 and anti-PD-1, flow cytometry was performed (Figure 4A). The population of NKp46+-activating NK cells was significantly increased in the combination group compared to the PBS group (Figure 4B). Next, the expression of IL12, IL15, and IFN-γ, which are known as pro-inflammatory and immune regulatory cytokines, was assessed at the mRNA level. The expression of IL12, IFN-γ, and IL15 was shown to significantly increase in the combination group compared to the PBS group (Figure 4C). These data indicate that the combination of CL7 and anti-PD-1 enhances the activating NK cells in the TNBC TME.

### 2.5. CL7 and Anti-PD-1 Suppressed the Lung Metastasis in the TNBC Model

To further evaluate the efficacy of the CL7 and anti-PD-1 combination treatment in suppressing lung metastasis in a TNBC model, lung tissues were stained with hematoxylin and eosin. Compared to the PBS group, lung nodules exhibited a significant reduction in the CL7 and anti-PD-1 monotherapy groups. In the combination treatment group, the lung nodule area was synergistically reduced compared to the PBS group (Figure 5A,B). Taken together, these findings suggest that CL7 and anti-PD-1 inhibit lung metastasis by reprogramming TAMs and activating CD8+ T cells and NK cells.

## 3. Discussion

Our study suggests that combining tumor-associated macrophage (TAM) reprogramming with immune checkpoint blockade is a potential treatment strategy for Triple-negative breast cancer (TNBC). The main finding of our research is that CL7, a CD300c antibody, and anti-PD-1 synergistically increased the M1/M2 ratio and activated CD8^+^ T cells and NK cells in the TNBC tumor microenvironment (TME), leading to a reduction in tumor volume. The potential clinical significance of our findings is that while neither TAM reprogramming nor immune checkpoint inhibition alone was sufficient, their combination successfully transformed the immune-suppressive TME into a pro-inflammatory environment, leading to a therapeutic effect.

TNBC accounts for 15–20% of breast cancer cases and has limited therapeutic strategies due to the lack of receptor targets [20]. Conventional treatment options for TNBC treatment include chemotherapy, radiation therapy, and surgery [21,22]. In recent years, immunotherapy using immune checkpoint inhibitors (ICIs), which enhance anti-cancer immune responses by targeting immunologic receptors on T cells, has demonstrated promising outcomes in various solid tumors, including advanced non-small cell lung cancer and renal cell carcinoma [23,24]. While most breast cancers are not inherently immunogenic, TNBC is considered the most immunogenic subtype of breast cancer for several reasons. First, TNBC exhibits a higher density of stromal and tumor-infiltrating lymphocytes. Second, although breast cancer generally has a lower tumor mutation burden (TMB) compared to other solid tumors, TNBC demonstrates a higher TMB than other breast cancer subtypes. Lastly, TNBC has been found to have higher rates of cell surface PD-L1 expression compared to other subtypes of breast cancer [25]. The FDA rapidly approved atetzolizumab in combination with a paclitaxel protein-bound agent in patients with locally advanced or metastatic TNBC with a PD-L1 expression rate greater than 1%. Subsequently, it approved pembrolizumab in combination with neoadjuvant chemotherapy in patients with high-risk, early-stage TNBC. Then, pembrolizumab was continued as a single agent in adjuvant treatment after surgery. Previous trials have shown positive results with pembrolizumab or atezolizumab treatment in TNBC. In the KEYNOTE-012 clinical trial (NCT01848834), pembrolizumab treatment for 27 PD-L1-positive TNBC patients showed an ORR of 18.5%, with a median response time of 17.9 weeks [26]. However, since most TNBC patients do not respond well to PD-1 or PD-L1 monotherapy, it seems particularly important to induce a favorable tumor immune microenvironment [27]. Therefore, there is a need for research on new strategies to increase the effectiveness of TNBC treatment through combination with ICI.

TAMs are the major component of TME, accounting for 30–50%, which is divided into M1 TAM and M2 TAM [28]. M1-like TAMs, activated by interferon-γ, toll-like receptors, lipopolysaccharide (LPS), and granulocyte macrophage colony-stimulating factor (GM-CSF), express NOS2 and IL-12B, and secrete pro-inflammatory cytokines such as TNF-α, IL-1, IL-6, and IL-12 [29,30]. M2-like TAMs, activated by IL-4 and IL-13, express Arg1 and CD206, and secrete anti-inflammatory cytokines like IL-10 and TGF-β, contributing to tissue repair and immune suppression [31]. TAM exhibits an immunosuppressive M2 phenotype in advanced cancer, which contributes to tumor growth, immunosuppression, invasion and migration, and metastasis. M2 TAMs express immune checkpoint ligands including PDL1, PDL2, B7-1, and B7-2, directly inhibiting T cell function [32]. Targeting IL-10+ TAMs can transform the immune-evasive microenvironment, presenting a potential therapeutic approach for gastric cancer [33]. In this respect, strategies that target TAM and use immune checkpoint inhibitors in combination have the potential to increase therapeutic effectiveness within TNBC.

CD300 molecules regulate immune responses through lipid-based ligands, with CD300c acting as an activator and CD300a as an inhibitor. CD300c is expressed on T cells, NK cells, macrophages, and neutrophils [34] and plays a role in modulating immune cell activation [35,36]. The activation of CD300c enhances NK cell degranulation and cytokine secretion, amplifying immune responses [37]. Recently, Lee et al. developed a CD300c-monoclonal antibody (CL7) that promotes M1 macrophage polarization by activating the MAPK and NF-κB signaling pathways. It was found that the CL7 antibody demonstrated therapeutic potential in the CT26 mouse model [16]. In this respect, treatment with anti-CD300c is considered to have the potential to activate immune cells. Based on previous studies, it was considered that there is a possibility of remodeling the TME through the combination of the CD300c antibody and anti-PD-1 within the TNBC model.

In our current study, we evaluated the novel therapeutic efficacy of a combination of CL7, which is a CD300c-monoclonal antibody, and anti-PD-1 in TNBC. Importantly, in our study, the combination of CL7 and anti-PD-1 collaboratively reduced tumor growth compared to the PBS group of the monotherapy group (Figure 1). A decrease in the number of infiltrating CD8^+^ T cells is associated with a poor prognosis for TNBC patients [38]. It has been reported that M2 TAMs inhibit the CD8^+^ T cell functions by impeding T cell proliferation and blocking T cell activation through interacting with the immune checkpoints [39]. Furthermore, targeting TAMs has the potential to affect other immune cells within the TME, thereby modulating an immune-stimulatory microenvironment. For example, targeting the scavenger receptor MARCO on TAMs alters TAM polarization and, in turn, activates natural killer (NK) cells to kill the tumor [40,41]. Consistent with this report, our study showed that CL7 and anti-PD-1 synergistically increased the M1/M2 ratio (Figure 2), which was accompanied by an enhancement in activated CD8^+^ T cells (Figure 3) and activated NK cells (Figure 4). While PD-L1 expression was not directly assessed in the current in vivo study, previous work demonstrated that CL7 treatment increased PD-L1 levels on THP-1 cells in vitro [16]. Together with the observed reduction in M2-like TAMs and the increase in CD8⁺ T cells and activated NK cells in the present study, these findings suggest that CL7 exerts immunomodulatory effects on the tumor microenvironment. This supports its potential role as a combination partner for immune checkpoint blockade therapy. Despite the higher PD-L1 expression and TMB in TNBC than in other breast cancers, it is often classified as a ‘cold tumor’, characterized by a lack of active immune cell infiltration and an immunosuppressive microenvironment [42,43]. As a result, monotherapy with anti-PD-1 alone may not be sufficient to elicit a strong immune response, necessitating combination therapies to boost the immune activity and improve therapeutic outcomes. In addition, TAM targeting therapies may have limited effectiveness due to various factors, including potential side effects and the diverse roles of TAMs across different cancers. This highlights the need for improved combination strategies with other therapies [32]. Compared to monotherapy, the underlying mechanism responsible for the observed synergistic effect in combination therapy remains unclear. Therefore, further studies on these findings are required.

Breast cancer metastasizes into the lungs, brain, skin, and bones, and 60–70% of breast cancer patients eventually die because of lung metastasis [44]. The loss of PI3K activity in tumor-associated macrophages (TAMs) promotes the development of pro-inflammatory mediators while reducing IL-10 and arginase expression, which contribute to immune suppression. PI3K inhibitors, in combination with immune checkpoint inhibitors, have demonstrated a synergistic effect in reducing tumor growth [45,46,47]. PI3K inhibition reprograms macrophages, alleviating the immunosuppressive state and limiting tumor cell proliferation and metastasis in breast cancer models [48]. There was a significant reduction in lung metastasis when using anti-PD-1 and CL7 monotherapy compared to in the PBS group, and more significant lung metastasis suppression was seen in the combination group (Figure 5A,B).

Despite these promising findings, this study has certain limitations. Notably, the absence of immunohistochemistry (IHC) or multiplex IHC data limits our ability to confirm the spatial distribution of immune cells such as TAMs, CD8⁺ T cells, and NK cells within the TME. While flow cytometry and RT-PCR were utilized to quantify immune cell populations and gene expression, spatial validation through IHC or spatial transcriptomic analysis would provide additional insights into the localization and functional interactions of the immune cells within metastatic lung tissues. Future studies incorporating such approaches would further validate and refine the proposed mechanism. Moreover, since this study was conducted in a murine model, the translatability of these findings to human patients remains uncertain. Additional preclinical and clinical studies will be required to assess the safety, efficacy, and therapeutic potential of this combination strategy across diverse patient populations.

Collectively, our results indicate that combination treatment with CL7 and anti-PD-1 could be a novel therapeutic option for suppressing breast cancer growth and metastasis by reprogramming the TME through an increase in M1 TAMs and the activation of CD8+ T cells and NK cells.

## 4. Materials and Methods

### 4.1. Reagents and Cell Lines

The clone CL7 was isolated from four rounds of biopanning against human CD300c using the synthetic human scFv library based on VH3–23 and VL1–47, with non-combinant complementarity determining region (CDR) diversity (unpublished results). InVivoMAb anti-mouse PD-1 (RMP1-14) was purchased from BioXCell (Lebanon, NH, USA). The 4T1 TNBC cell line purchased from ATCC was cultured in RPMI1640 media containing 10% fetal bovine serum (FBS; Welgene, Gyeongsan, Republic of Korea), 100 U/mL penicillin, and 100 μg/mL streptomycin (Gibco, Thermo Fisher Scientific, Waltham, MA, USA). The cells were cultured every 2–3 days until 80% confluence, and were incubated at 37 °C in a humidified 5% CO_2_ incubator.

### 4.2. Animal Study

Female BALB/c wild-type mice were purchased from DBL (Chungbuk, Republic of Korea). Animal procedures were approved by the University of Kyung Hee Institutional Animal Care and Usage Committee (KHUASP(SE)-24-041). All animals were maintained in a pathogen-free environment with a 12 h light/dark cycle and were supplied with water and food ad libitum. To generate the mouse TNBC model, 4T1 cells were mixed with Matrigel (Corning, NY, USA). Cells were inoculated with 1 × 10^5^ cells per mouse in the left 4th mammary gland of the mice (6–8 weeks old). Seven days after inoculation, mice were intravenously injected with CL7 (5 and 10 mg/kg, intraperitoneal injection; 10 and 30 mg/kg, intravenous injection (CentricsBio; Seoul, Republic of Korea)) or anti-PD-1 antibody (200 μg; BioXcell) twice a week. The dosage and administration routes for CL7 were determined based on preliminary dos-response experiments and a previous paper demonstrating effective macrophage modulation without significant toxicity [16]. Intravenous (IV) and intraperitoneal (IP) routes were used to compare the systemic and peritoneal delivery efficiencies. The dosing schedule (twice a week for five doses) was selected to allow sustained immune modulation while minimizing adverse effects. Before each injection, the tumor size and body weight were measured using a digital caliper and a digital scale, respectively. The tumor volume was determined using the following equation: (width × width × length)/2. Mice were euthanized after five injections were administered. Mice were euthanized according to the guidelines when the tumor diameter reached 2 cm. Tissue samples were collected immediately after euthanasia for analysis.

### 4.3. Flow Cytometry

Tumor tissues were harvested from the sacrificed mice and put in the MACS C tube (Miltenyi Biotec, Auburn, CA, USA) containing Collagenase D (1 mg/mL; Sigma-Aldrich, St. Louis, MO, USA) and DNase1 (1 mg/mL; Sigma-Aldrich) in serum-free medium. The tissues were dissociated using a MACS dissociator (Miltenyi Biotec) and digested for 25 min at 37 °C with a shaking incubator. The tissues were then filtered using a 40 μm cell strainer (Corning Incorporated, Corning, NY, USA) to obtain a single-cell suspension. Red blood cells (RBCs) were lysed with 1× RBC lysis buffer (Invitrogen, Carlsbad, CA, USA) for 5 min at room temperature. Cells were washed and resuspended in BD Pharmingen™ Stain Buffer (BD bioscience, San Jose, CA, USA). The cells were stained for 45 min at 4 °C with antibodies.

The following antibodies were used: For the identification of monocytes/macrophages, we used mouse CD45-FITC (Biolegend, San Diego, CA, USA), CD11b-BV510 (BD bioscience), CD86-PE-Cy7 (Biolegend), and CD206-APC (Biolegend); For the identification of CD8^+^ T cells, we used mouse CD45-APC-Cy7 (BD), CD8-PE-Cy7 (BD), CD4-BB700 (BD), and Granzyme B-APC (Invitrogen); For the identification of activating NK cells, we used mouse CD45-APC (Biolegend), CD3ε-FITC (Biolegend), and NKp46-BV711 (Biolegend).

For intracellular staining, the cells were treated with 1x fixation and permeabilization buffer (BD Biosciences) for 30 min. The single-cell suspension was washed and stained with Granzyme B. The data were acquired using a BD FACSlyric™ (BD Biosciences) flow cytometry system and analyzed using the BD FACSuite software 1.2.1 (BD Biosciences).

### 4.4. Quantitative Real-Time PCR (qRT-PCR)

Total RNA was isolated from tumor tissues using an easy-BLUE RNA extraction kit (iNtRON Biotechnology, Seongnam, Republic of Korea). cDNA was synthesized using Cyclescript reverse transcriptase (Bioneer, Daejeon, Republic of Korea) according to the manufacturer’s instructions. Real-time PCR was performed using a CFX connect real-time PCR system (Bio-Rad La-boratories, Hercules, CA, USA) and the SensiFAST SYBR no-Rox kit (Bioline, London, UK). The expression levels of the target mRNAs were normalized to the expression levels of mouse GAPDH, a housekeeping gene. All fold-changes were expressed relative to the PBS group. Each reaction was performed in duplicates. The base sequences of the primers used are listed in Table 1.

### 4.5. Hematoxylin and Eosin Staining

Lung tissues from tumor-bearing mice were fixed in 10% formalin. Lung tissue was embedded in paraffin and sectioned at 5 μm thickness. Deparaffinization was performed on the tissue sections with xylene for 10 min. The sections were dehydrated and stained with hematoxylin (Cancer Diagnostics, Inc., Durham, NC, USA) for 1 min and washed with tap water for 10 min. The slides were then stained with eosin for 30 s and washed again in tap water for 10 min, and dehydrated with increasing concentrations of ethanol. Tissue slides were mounted and photographed at 1.25× under a light microscope (Olympus, Tokyo, Japan). The lung nodule area was calculated using ImageJ software (1.54d version).

### 4.6. Statistics

The data collected were analyzed using Prism 5.01 software (GraphPad Software Inc., San Diego, CA, USA), and are expressed as the mean ± standard error of mean (SEM). All data were tested for normality using the normality test in GraphPad prism. One-way analysis of variance (ANOVA) followed by the Newman–Keuls test or two-way ANOVA followed by the Bonferroni post hoc test was performed for group comparisons. *p* < 0.05 was considered to indicate a statistically significant difference.

## Figures and Tables

**Figure 1 ijms-26-05045-f001:**
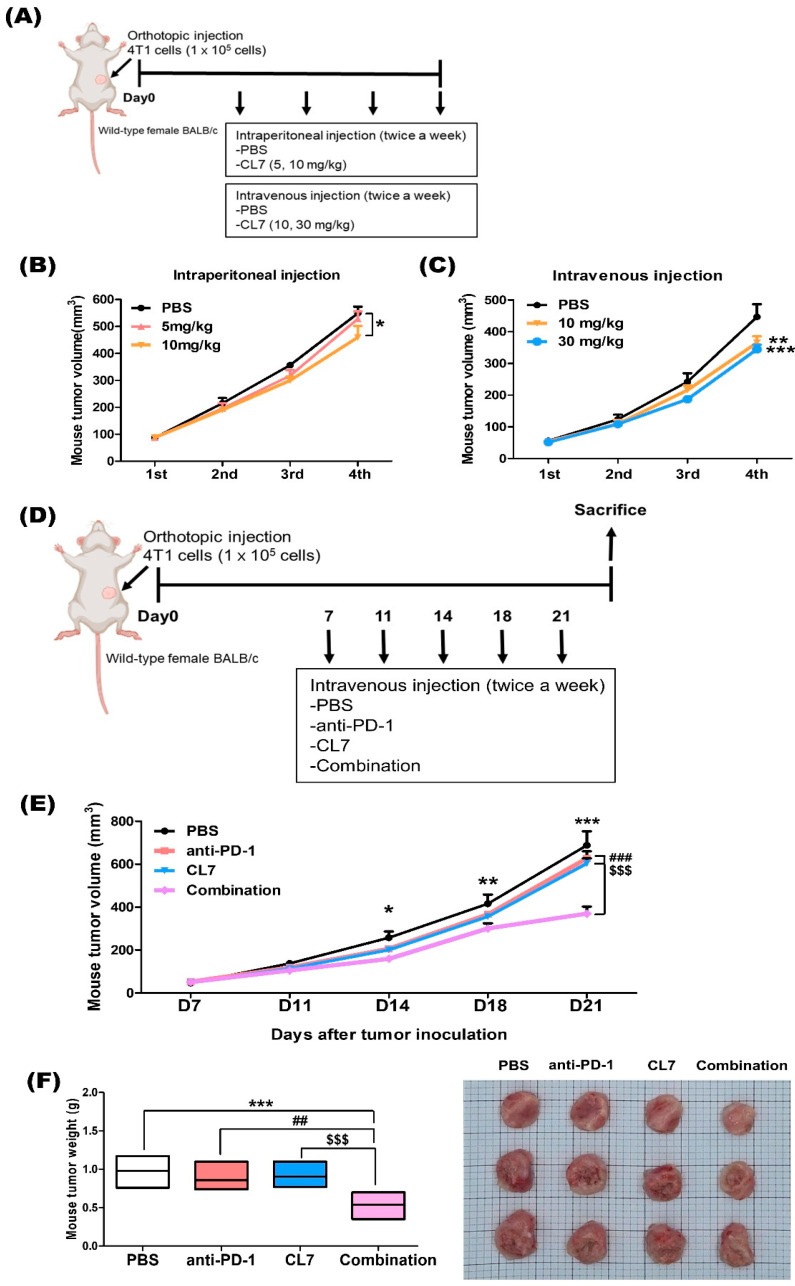
Combination treatment of CL7 and anti-PD-1 synergistically reduced the tumor growth in 4T1 TNBC model. (**A**) In vivo experimental schedule. 4T1 cells were inoculated with 1 × 10^5^ cells per mouse in the left 4th mammary gland of the mice (6–8 weeks old). Seven days after inoculation, mice were (**B**) intraperitoneally injected with CL7 (5 mg/kg, 10 mg/kg) or (**C**) intravenously injected with CL7 (10 mg/kg, 30 mg/kg) twice a week. After four rounds of administration, mice were euthanized. All data are presented as the mean ± SEM; * *p* < 0.05, ** *p* < 0.01, *** *p* < 0.001 versus PBS group (*n* = 4). (**D**) 4T1 cells were inoculated described as above. Seven days after inoculation, mice were intravenously injected with CL7 (30 mg/kg) or anti–PD-1 antibody (200 μg) twice a week. (**E**) The volume of the tumor was measured using a digital caliper. Tumor volume was calculated using the following equation: (width × width × length)/2. (**F**) After five rounds of administration, mice were euthanized. Tumor tissues were imaged, and the tumor weight was measured using an electronic scale. All data are presented as the mean ± SEM; * *p* < 0.05, ** *p* < 0.01, *** *p* < 0.001 versus PBS group; ## *p* < 0.01, ### *p* < 0.001 versus anti-PD-1 group; and $$$ *p* < 0.001 versus CL7 group (*n* = 6).

**Figure 2 ijms-26-05045-f002:**
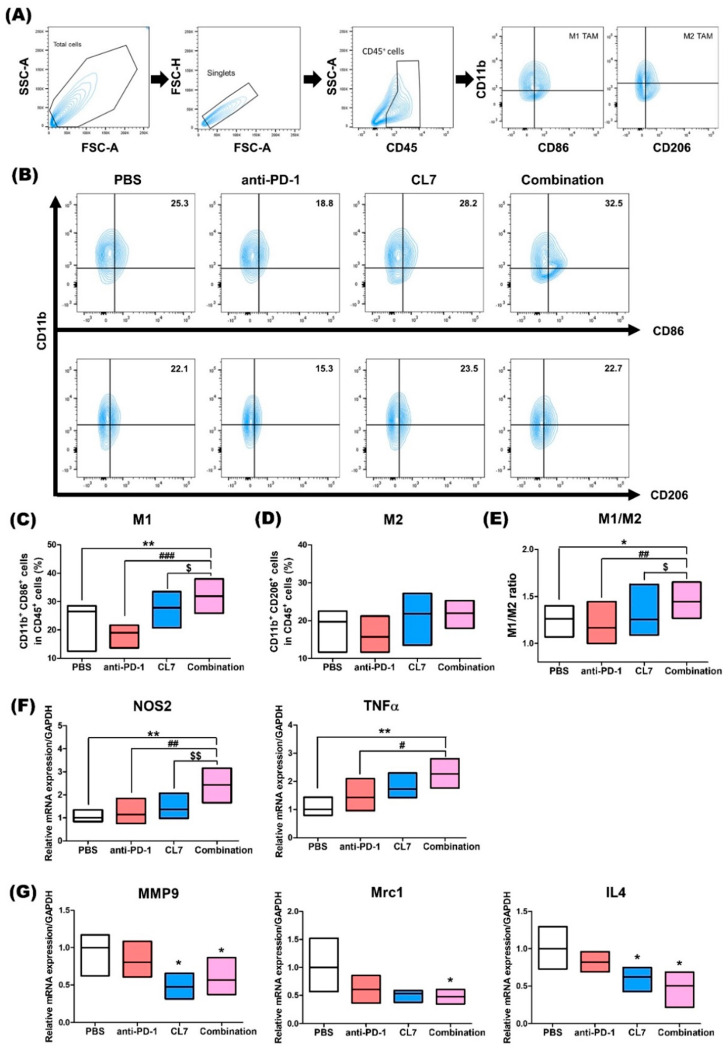
Combination treatment of CL7 and anti-PD-1 has effects on the macrophage population in the tumor microenvironment of TNBC. (**A**) Identification of myeloid cell populations following exclusion of doublets. CD45^+^ gate was also used as a first step for specific immune cell identification. CD45^+^CD11b^+^CD86^+^ cells were regarded as M1 macrophages, whereas CD45^+^CD11b^+^CD206^+^ cells were regarded as M2 macrophages. (**B**–**D**) Percentages of the M1 and M2 macrophage populations. (**E**) The M1/M2 ratio was calculated by dividing the M2 population by the M1 population (*n* = 8). After the in vivo experiment, tumor tissues were harvested and RNA was extracted for the analysis (*n* = 4). The mRNA expression of (**F**) *NOS2, TNFα* (M1 macrophage-related genes), (**G**) *MMP9, Mrc1,* and *IL4* (M2 macrophage-related genes) was measured using RT-PCR. All data are presented as the mean ± SEM; * *p* < 0.05, ** *p* < 0.01 versus PBS group; # *p* < 0.05, ## *p* < 0.01, ### *p* < 0.001 versus anti-PD-1 group; and $ *p* < 0.05, $$ *p* < 0.01 versus CL7 group.

**Figure 3 ijms-26-05045-f003:**
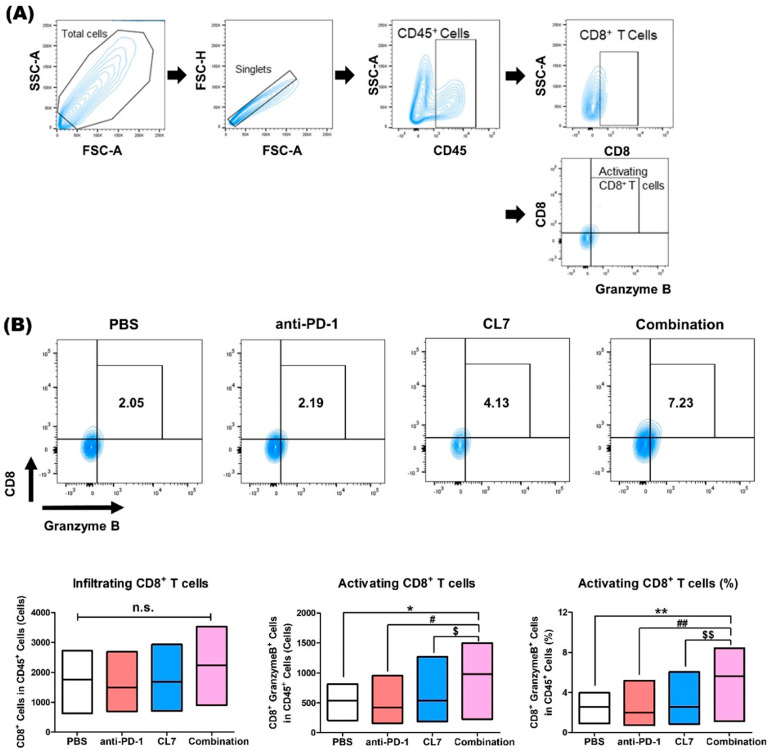
Combination treatment of CL7 and anti-PD-1 significantly enhanced the activating CD8+ T cells in the breast cancer tissue. (**A**,**B**) Identification of lymphoid cell populations following exclusion of doublets. CD45+ gate was also used as a first step for specific immune cell identification. CD45+CD8+ cells were considered infiltrating CD8+ T cells, and CD45+CD8+GranzymeB+ cells were considered infiltrating activated CD8+ T cells. All data are presented as the mean ± SEM; * *p* < 0.05, ** *p* < 0.01 versus PBS group; # *p* < 0.05, ## *p* < 0.01 versus anti-PD-1 group; and $ *p* < 0.05, $$ *p* < 0.01 versus CL7 group. n.s., not significant (*n* = 8).

**Figure 4 ijms-26-05045-f004:**
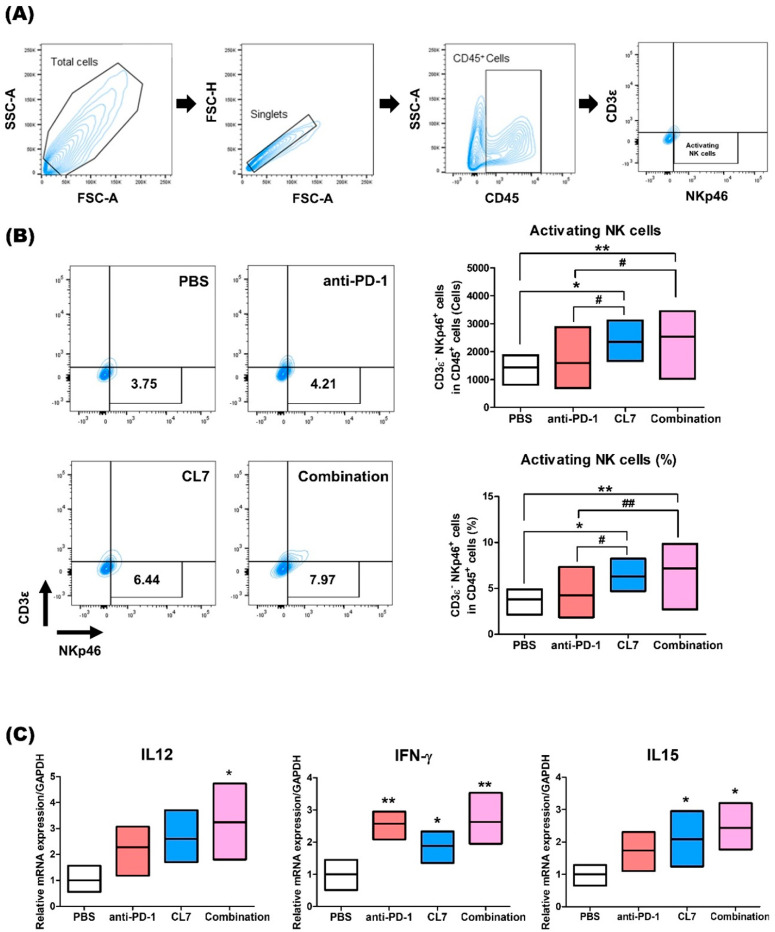
CL7 and anti-PD-1 combination treatment promotes NK cell activation in breast cancer. (**A**) Identification of NK cell populations following exclusion of doublets. CD45+ gate was also used as a first step for specific immune cell identification. CD45+CD3ε-NKp46+ cells were regarded as activating NK cells. (**B**) The number of activated NK cell populations (upper panels) and the percentage of activated NK cells (lower panels) (*n* = 8). (**C**) The expression of *IL12, IFN-γ*, and *IL15* was assessed at the mRNA level (*n* = 4). All data are presented as the mean ± SEM; * *p* < 0.05, ** *p* < 0.01 versus PBS group; # *p* < 0.05, ## *p* < 0.01 versus anti-PD-1 group.

**Figure 5 ijms-26-05045-f005:**
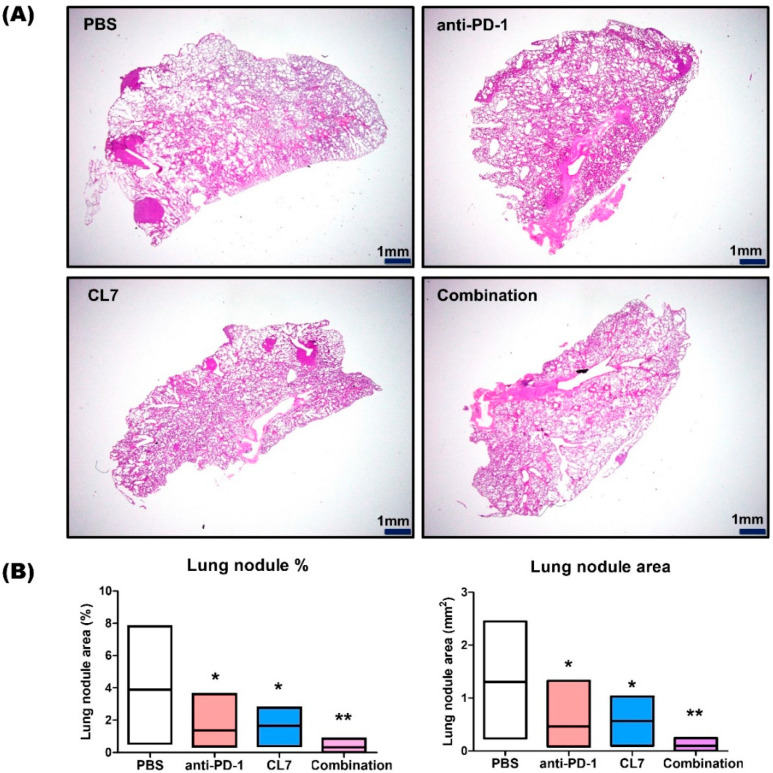
CL7 and anti-PD-1 suppressed lung metastasis in the TNBC Model. (**A**) Image of H&E-stained lung tissue. Magnification, 1.25×. Scale bar, 1 mm. (**B**) Lung nodule area was calculated using Image J (*n* = 7). All data are presented as the mean ± SEM; * *p* < 0.05, ** *p* < 0.01 versus the PBS group.

**Table 1 ijms-26-05045-t001:** Primers used in real-time PCR.

Gene	F/R	Primer Sequence
*GAPDH*	F	5′-ACC CAG AAG ACT GTG GAT GG-3′
	R	5′-CAC ATT GGG GGT AGG AAC AC-3′
*Mrc1*	F	5′-TTC GGT GGA CTG TGG ACG AGC-3′
	R	5′-ATA AGC CAC CTG CCA CTC CGG-3′
*IL-4*	F	5′-ATC CTG CTC TTC TTT CTC GAA TGT-3′
	R	5′-GCC GAT GAT CTC TCT CAA GTG ATT-3′
*MMP9*	F	5′-TGA ATC AGC TGG CTT TTG TG-3′
	R	5′-ACC TTC CAG TAG GGG CAA CT-3′
*NOS2*	F	5′-GGC AGC CTG TGA GAC CTT TG-3′
	R	5′-GAA GCG TTT CGG GAT CTG AA-3′
*TNFα*	F	5′-CAT CTT CTC AAA ATT CGA GTG ACA A-3′
	R	5′-TGG GAG TAG ACA AGG TAC AAC CC-3′
*IFN-γ*	F	5′-TCA AGT GGC ATA GAT GTG GAA GAA-3′
	R	5′-TGG CTC TGC AGG ATT TTC ATG-3′
*IL12*	F	5′-AAG CTC TGC ATC CTG CTT CAC-3′
	R	5′-GAT AGC CCA TCA CCC TGT TGA-3′
*IL15*	F	5′-GAT TGA AGG GAA GCA ACG GG-3′
	R	5′-GCA CTC TCC AAC CCA CTT GA-3′

## Data Availability

All data generated or analyzed during this study are included in this published article.

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
