# Peer review of "The Combination of CD300c Antibody with PD-1 Blockade Suppresses Tumor Growth and Metastasis by Remodeling the Tumor Microenvironment in Triple-Negative Breast Cancer"

_ijms, 2025, doi:10.3390/ijms26115045_

Round 1

Reviewer 1 Report

Comments and Suggestions for Authors

This study proposed that combining tumor associated macrophage (TAM) reprogramming with immune checkpoint blockade is a potential treatment strategy for triple-negative breast cancer (TNBC) through modulation of M1/M2 ratio, activating CD8-T cell, and NK cells. I have 2 questions to the study.

  1. Did you analyze the tumor section with IHC or multiplex IHC staining of the macrophage/CD8+T cell/NK distribution within tumors?
  2. In previous paper (Immunobiology. 2024 Jan;229(1):152780), did you have similar finding of increased PD-L1 expression after CD300c treatment?

Author Response

Thank you for your valuable comments.
We have revised the manuscript accordingly.
Please see the attached revised manuscript for details.

Reviewer 2 Report

Comments and Suggestions for Authors

Dear authors:

This well-designed study demonstrates promising efficacy of combined CL7 (CD300c antibody) and anti-PD-1 therapy in TNBC by remodeling the tumor microenvironment through increased M1 macrophages and activated CD8+ T/NK cells. While the methodology is robust and results clearly presented, the manuscript would benefit from: (1) stronger mechanistic rationale in the introduction for targeting CD300c specifically; (2) more detailed justification of experimental parameters (doses, timing, sample sizes); and (3) brief discussion of limitations (e.g., translational challenges). These revisions would enhance the paper's impact without requiring additional experiments. The study's novel approach and compelling data make it suitable for publication after addressing these points.

Author Response

(The authors gave the same response as above.)
